# QAID: QUESTION ANSWERING INSPIRED FEW-SHOT INTENT DETECTION

**Asaf Yehudai** [◇♣] **, Matan Vetzler** [◇]**, Yosi Mass** [◇]**, Koren Lazar** [◇]**, Doron Cohen** [◇]**, Boaz Carmeli** [◇]

IBM Israel Research Lab [◇], Hebrew University of Jerusalem [♣]
{first.last}@ibm.com, {yosimass, doronc, boazc}.il.ibm.com

## ABSTRACT

Intent detection with semantically similar fine-grained intents is a challenging task. To address it, we reformulate intent detection as a question-answering retrieval task by treating utterances and intent names as questions and answers. To that end, we utilize a question-answering retrieval architecture and adopt a two stages training schema with batch contrastive loss. In the pre-training stage, we improve query representations through self-supervised training. Then, in the fine-tuning stage, we increase contextualized token-level similarity scores between queries and answers from the same intent. Our results on three few-shot intent detection benchmarks achieve state-of-the-art performance.

## 1 INTRODUCTION

Intent detection (ID) is the task of classifying an incoming user query to one class from a set of mutually-exclusive classes, a.k.a. intents (Wang et al., 2014; Schuurmans & Frasincar, 2019; Liu et al., 2019a). This ability is a cornerstone for task-oriented dialogue systems as correctly identifying the user intent at the beginning of an interaction is crucial to its success. However, labeled data is required for training and manual annotation is costly. This calls for sample efficient methods, gaining high accuracy with minimal amounts of labeled data.

Recent works tackling few-shot ID have relied on large-scale pre-trained language models, such as BERT (Devlin et al., 2018). These works leverage task-adaptive training and focus on pre-training a model on a large open-domain dialogue corpus and fine-tuning it for ID classification (Mehri et al., 2020; Wu et al., 2020a; Casanueva et al., 2020; Zhang et al., 2021a).

Alternative approaches tried to learn query representation based on query-to-query matching (henceforth, Match-QQ systems) (Zhang et al., 2020; Mass et al., 2020; Mehri et al., 2021). Zhang et al. (2020); Mass et al. (2020) adopt pairwise-encoding systems with cross-attention to deploy K-Nearest-Neighbor (K-NN) (Fix & Hodges, 1989) classification schema where training queries are fully utilized for both training and inference stages. Nevertheless, those methods' downside is the processing time combined with the difficulty of scaling to large number of intents (Liu et al., 2021c).

The need to efficiently compare an incoming query to a large set of possible answers resides at the core of any question answering (QA) retrieval system (henceforth, Match-QA systems) (Karpukhin et al., 2020). Recently, Khattab & Zaharia (2020) introduced ColBERT, which allows faster training and inference by replacing the cross-attention mechanism used by Match-QQ systems (Zhang et al., 2020; Mass et al., 2020; Nogueira & Cho, 2019) with a fast contextualized token-level similarity mechanism dubbed late interaction.

In this work, we present a Question Answering inspired Intent Detection system, named QAID. We start by formulating the ID task as a question-answering retrieval task by treating the utterances and the intent names as queries and answers, respectively. This reformulation allows us to introduce valuable additional signal from the intent names. Then, we adapts the efficient architecture of Col-BERT while replacing its triplet function loss with batch contrastive loss which was proven to be more robust (Khosla et al., 2020) and performs well in various tasks (Gunel et al., 2021; Gao et al., 2021a), including ID classification (Zhang et al., 2021b). In contrast to ColBERT which compares a query to a pair of positive and negative documents, we also include queries as positive examples,

and so we compare the queries both to their answers and to other queries from the same intent. This allows QAID to represent similarly both queries and answers of the same intent. Therefore, our training method assumes the settings of both Match-QQ and Match-QA. In inference, QAID relies on the token-level similarity (late interaction) mechanism between incoming query and all intent names for its predictions (Khattab & Zaharia, 2020).

Our contribution is thus threefold. (1) We show that few-shot intent detection can be successfully handled by QA systems when letting the intent name play the role of the answer. (2) We show how intent detection architectures can benefit from recent advancements in supervised batch contrastive training and late-interaction scores. (3) We report state-of-the-art results on three few-shot intent detection benchmarks.

## 2    METHOD

Our method addresses the few-shot intent detection task, in which we have $C$ defined intents and the task is to classify an incoming user query, $q$, into one of the $C$ classes. In our formulation, upon getting a new user query $q$, we need to retrieve the most suited intent name. We set balanced K-shot learning for each intent (Mehri et al., 2020; Casanueva et al., 2020; Zhang et al., 2020), i.e., the training data containing K examples per intent[1].

In the following section, we describe the structure of our QAID framework and its training stages. First, in Section 2.1 we elaborate on the different components of QAID. Then, in Section 2.2 we present the two training stages: the self-supervised contrastive pre-training in 2.2.1 and the supervised batch contrastive fine-tuning in 2.2.2. Lastly, in Section 2.3 we briefly touch on our decision to formulate ID as a question retrieval task.

### 2.1    REPRESENTATION LEARNING FRAMEWORK

The main components of our framework are:

- **Data Augmentation module,** $Aug(\cdot)$. For each input query, $q$, we generate two random augmentations, $\hat{q} = Aug(q)$, each of which represents a different view of the input, $q$. For our augmentation we use the combination of two simple and intuitive 'corruption' techniques (Gao et al., 2021a; Wu et al., 2020b; Liu et al., 2021b); (i) randomly masking tokens from $q$ (Devlin et al., 2018); (ii) dropping a small subset of neurons and representation dimensions. Technique (i) is done before passing the query to the encoder and technique (ii) is done in the forward propagation through the encoder model.

- **Encoder model,** $Enc(\cdot)$, which maps a query $q$, consisting $q_1, ..., q_m$ tokens, to $Enc(q) \in R^{m \times D_E}$, where $D_E$ is the embedding dimension. In our experiments, it is either 768 or 1024.

- **Projection layer,** $Proj(\cdot)$, a single linear layer that maps vectors of dimension $D_E$ to vectors of dimension $D_P = 128$, followed by normalization to the unit hypersphere.

- **Token-level score,** $Score(\cdot, \cdot)$, given two queries $u = (u_1, ..., u_m)$ and $v = (v_1, ..., v_l)$, the relevance score of $u$ regarding to $v$, denoted by $Score(u, v)$, is calculated by the late interaction between their bags of projected contextualized representations, i.e $z(u) = Proj(Enc(u))$. Namely, the sum of the maximum token-wise cosine similarity of their projected representations (Khattab & Zaharia, 2020). Equation 1 shows the formulation of this score.

$$Score(u, v) = \sum_{i \in [m]} \max_{j \in [l]} z(u)_i \cdot z(v)_j \tag{1}$$

### 2.2    TWO-STAGE CONTRASTIVE TRAINING

In both stages, given a batch of input samples $Q = (q^1, ..., q^n)$, we first apply $Aug$ followed by the encoding and projection layer, denoted by the $z(\cdot)$ function as described in the last section, and so we have $X_Q = z(Aug(Q)) \in R^{2n \times D_P}$, where the $2n$ is a result of the two random augmentations we applied to each query. In the self-supervised training, each two augmented queries are the only

---

[1]In the rest of the paper we refer to the intent examples as queries and use intents and classes interchangeably

positive examples for each other while all other queries are negative. In the supervised training phase, we also run the same process with the same encoder on the corresponding intent names, $A = (a^1, ..., a^n)$, resulting in $X_A = z(Aug(A)) \in R^{2n \times D_P}$. $X_A$ together with $X_Q$ forms a training batch of $4n$ instances. In this supervised setting, all queries and intent names of the same intent are positive to each other while all others are negative.

### 2.2.1 PRE-TRAINING

We use a task-adaptive pre-training stage to overcome the few-shot constraint, as done by most goal-oriented dialogue works (Mehri et al., 2020; Zhang et al., 2020). Our pre-training aims to facilitate domain adaptation by two early-stage objectives: (1) Incorporate token-level domain knowledge into the model (Mehri et al., 2020; 2021); (2) Adopt queries' representations to the dialog domain through data augmentation techniques and self-supervised contrastive training (Zhang et al., 2021b; 2022a).

Practically, in a training batch containing $2n$ augmented queries, let $t \in [2n]$ be the index of an arbitrary augmented query. Then in the self-supervised contrastive learning stage, the loss takes the following form:

$$\mathcal{L}^{self} = - \sum_{t \in [2n]} log \frac{exp(Score(q^t, q^{J(t)})/\tau)}{\sum_{a \in A(t)} exp(Score(q^t, q^a)/\tau)} \tag{2}$$

where t is called the anchor/pivot, $J(t)$ is the index of the second augmented sample deriving from the same source sample (a.k.a positive), $A(t) = \{[2n] \setminus t\}$, $A(t) \setminus J(t)$ are called the negative and $\tau \in R^+$ is a scalar temperature parameter that controls the penalty to negative queries (see step (a) in Figure 1).

**Masked language modeling as an auxiliary loss**: In addition to the self-supervised contrastive training, we pre-train also on the masked language modeling (MLM) task (Taylor, 1953), to further adjust sentence-level representation to the domain of the data. Moreover, this improves lexical-level representation which is essential for token-level similarity. Hence we define $\mathcal{L}^{mlm}$ as the average cross-entropy loss over all masked tokens.

**The overall loss** for the pre-training phase is $\mathcal{L}^{PT} = \mathcal{L}^{self} + \lambda \mathcal{L}^{mlm}$, where $\lambda$ is a controllable hyperparameter.

### 2.2.2 FINE-TUNING

At the fine-tuning stage, we only have a limited number of examples for each intent, and intents may be semantically similar, making the classification task difficult. To address the data scarcity, we utilize the explicit intent names as unique examples that serve as answers in our QA retrieval framework. This is similar to recent works that leverage label names for zero-shot or few-shot text classification (Meng et al., 2020; Basile et al., 2021). The intent name is usually assigned by a domain expert when designing a goal-oriented dialogue system. As such, it provides a semantic description of the intent that aims to discriminate it from other intents. Consequently, intent names may provide a signal that is otherwise difficult to extract from a small set of queries.

Additionally, we utilize the pre-trained model designed for queries' representations and continue fine-tuning it on the few-shot examples with the permanent representation learning technique (Khosla et al., 2020) of supervised batch contrastive training (see step (b) in Fig. 1). In that way, each time our model pulls together two queries from the same class it simultaneously also pulls their intent names closer together as well. Formally our supervised contrastive loss has the form:

$$\mathcal{L}^{sup} = \sum_{t \in [4n]} \frac{-1}{|P(t)|} \sum_{p \in P(t)} log \frac{exp(Score(q^t, q^p)/\tau)}{\sum_{a \in A(t)} exp(Score(q^t, q^a)/\tau)} \tag{3}$$

In this formulation $q$ may represent either an augmented query or its intent name, and since each instance has four views: two augmented queries and two augmented intent names, we have a total

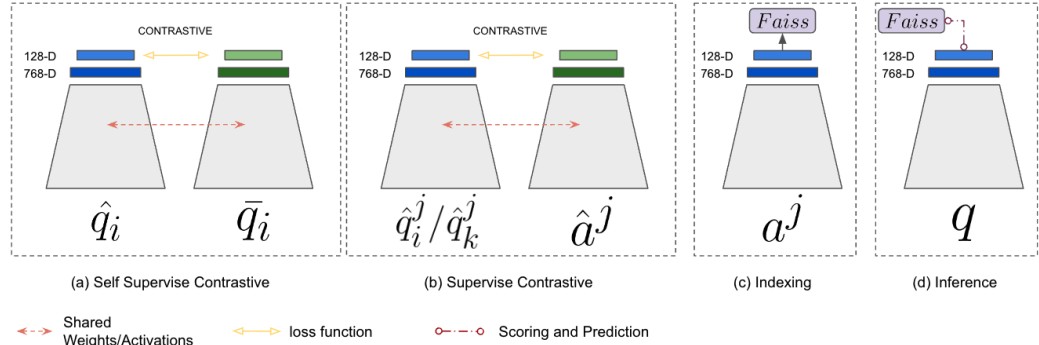

Figure 1: Schematic illustration of our method. Hat and bar represent augmentation, the subscript is a ruining index, superscript is the class index. (a) Self-supervised contrastive training with data augmentation to enhance quires' representations. (b) Supervise contrastive fine-tuning to learn query-to-query and query-to-answer similarity. (c) Indexing the answers into Faiss index. (d) Compare incoming query against all answers in the index and predict the most similar one.

of $4n$ samples. $A(t) = \{[4n] \setminus t\}$; $P(t)$ is the group of all samples that are positive to $q_t$ and is defined as all the augmented queries or intent names derived from the same label as $q_t$.

Besides the supervised contrastive loss we also train with a classification loss, $\mathcal{L}^{clss}$, and an MLM loss, $\mathcal{L}^{mlm}$. In total, the fine-tuning loss is $\mathcal{L}^{FT} = \mathcal{L}^{sup} + \lambda_{class}\mathcal{L}^{class} + \lambda_{mlm}\mathcal{L}^{mlm}$, where $\lambda_{class}$ and $\lambda_{mlm}$ are controllable hyperparameters.

**Indexing and Inference.** After fine-tuning, we index the embeddings of all candidate answers (a.k.a intent names) outputted from the encoder into Faiss (Johnson et al., 2021), a library for large-scale vector-similarity search. see Khattab & Zaharia (2020) for more details. Then, at inference time, we compare the incoming query representation with all answers in the index and retrieve the most similar (see step (c) and (d) in Fig. 1).

## 2.3 WHY TO FORMULATE THE TASK AS A QUESTION ANSWERING RETRIEVAL TASK?

For a few-shot intent detection model to be practical, it must be computationally efficient in both training and inference phases, while possibly handling a large number of intents (Qi et al., 2020). Here we are leveraging the recent success of dense passage retrieval for question answering that was shown to perform well and efficiently (Karpukhin et al., 2020; Khattab & Zaharia, 2020). Those systems can handle a large number of candidate answers by using (1) a dual-encoder framework with light comparison instead of the computationally demanding pairwise-encoding, (2) an indexation that enables a large-scale fast retrieval.

## 3 EXPERIMENTAL SETUP

### 3.1 DATASETS

We experiment with three widely studied few-shot intent detection datasets which represent the intent detection (ID) part of DialoGLUE benchmark.[2] These datasets present challenging few-shot ID tasks with fine-grained intents that are semantically similar. Moreover, they facilitate the comparison with recent state-of-the-art baselines.

**Clinc150** (Larson et al., 2019) contains 22,500 personal assistance queries classified into 150 intents across 10 domains.
**Banking77** (Casanueva et al., 2020) contains 13,242 online banking queries classified into 77 fine-grained intents in a single domain.

---

[2]For readily use: https://github.com/jianguoz/Few-Shot-Intent-Detection/tree/main/Datasets

**HWU64** (Liu et al., 2019b) contains 11,106 personal assistant queries classified into 64 intents across 21 different domains. Table 1 reports the data splits of each dataset.

| Dataset | #Train | #Vaild | #Test | #Intents | #Domains |
|---|---|---|---|---|---|
| CLINC150 | 15000 | 3000 | 4500 | 150 | 10 |
| BANKING77 | 8622 | 1540 | 3080 | 77 | 1 |
| HWU64 | 8954 | 1076 | 1076 | 64 | 21 |

Table 1: Data statistics of the three intent detection datasets from DialoGLUE.

## 3.2 MODELS AND BASELINES

We experiment with RoBERTa-base and RoBERTa-large encoders from the Hugging Face transformers library[3] with ColBERT architecture.[4] In the self-supervised pre-training stage, we utilize the training and validation sets of the six ID datasets from Zhang et al. (2021b); Mehri et al. (2020; 2021). For a fair evaluation, we exclude the test sets from our pre-training following the observation of Zhang et al. (2021b). We fix the number of embeddings per query at $m = 32$ same as Khattab & Zaharia (2020). For the MLM loss, we follow the masking strategy of Devlin et al. (2018). We also use this masking strategy to augment our input queries in addition to the representation-level augmentation through the encoder build in $10\%$ dropout. For the contrastive loss, we use the implementation of Khosla et al. (2020)[5]. For the late interaction score, we normalize the score by the number of tokens in the summation. We train our encoder for 20 epochs with a batch size of 64, a learning rate of $1e^{-5}$, a temperature parameter $\tau$ of 0.07 (same as Khosla et al. (2020)) and $\lambda = 0.1$ as recommended in the literature.

For the fine-tuning stage, we train our models on 5- and 10-shot splits as available within the public dataset distribution. We train our model for 10 epochs starting from the pre-trained model checkpoint. We set the batch size to 32, where queries and answers are encoded separately by the same encoder. Answers are truncated by the longest one in the batch same as Khattab & Zaharia (2020). We apply the same masking schema to queries and answers as done in the pre-training. We set the temperature to 0.07. We also set $\lambda_{class}$ and $\lambda_{mlm}$ to 0.1 and 0.05, respectively, making $\mathcal{L}^{sup}$ the main summand in $\mathcal{L}^{FT}$. We used those parameters as they were recommended in the literature and shown to perform best in hyperparameter tuning. Following previous works, we ran the experiments with five different seeds and report the average accuracy.

For our Faiss Index[6] implementation, we use IVFScalarQuantizer ("InVerted File with Scalar Quantizer."). To improve memory efficiency, every embedding is represented by 8 bytes. In our work, we use the full retrieval option that effectively retrieves all candidate answers. In cases of many intents, one can deploy fast retrieval of top candidate answers.

## 3.3 BASELINES

We start by categorizing the baselines by three main representation and prediction methodologies. A **Classifier** architecture learns both query representation and a linear classification head via cross-entropy loss. The classification head contains a single vector per class that is implicitly trained to represent the class and enable prediction at inference time. A **Match-QQ** architecture learns query representation via query-to-query similarity matching as it learns to increase the similarity between embeddings of queries from the same class while simultaneously decreasing the similarity between queries from disparate classes. During inference, an input query is matched against a large collection of training queries, and the intent in which its queries are the most similar to the input is predicted. A **Match-QA** architecture learns to match queries to answers. The model learns to increase the similarity between queries and their corresponding answers while simultaneously decreasing the similarity

---

[3]https://github.com/huggingface/transformers

[4]https://github.com/stanford-futuredata/ColBERT/tree/colbertv1

[5]https://github.com/HobbitLong/SupContrast/blob/master/losses.py

[6]https://github.com/facebookresearch/faiss

between queries and other answers. At inference time, an incoming query is matched against all possible answers, and the most similar is predicted. In these terms, the pretraining of QAID is based on Match-QQ, its fine-tuning involves both Match-QQ and Match-QA, and its prediction is based on Match-QA. We experimented with prediction methods that include also Match-QQ, but found it preform slightly worse than inference with only Match-QA. We will elaborate more in Section §4.1.

### 3.3.1 BASELINE MODELS

We compare our approach and results against strong baseline models reported in the literature. In the rest of the section, we discuss these models in more detail and align them with the paradigms mentioned above. Notably, some baseline models mix and match components across architectures.

- **Classifier**: We fine-tune RoBERTa-base encoder with a feed-forward classification head as our classifier baseline.

- **ColBERT** (Khattab & Zaharia, 2020): Contextualized Late Interaction over BERT (ColBERT) is a state-of-the-art passage search and retrieval system. ColBERT provides a Match-QA baseline and is the basis for the QAID architecture. For training we use 20 triplets (query, pos answer, neg answer) for each query, with hard negatives, namely, we run the query against all answers using bm25 (Robertson & Zaragoza, 2009) and select the negatives from the most similar answers.

- **USE+CONVERT** (Casanueva et al., 2020): USE (Yang et al., 2019) is a large multilingual dual-encoder model pre-trained in 16 languages. CONVERT (Casanueva et al., 2020) is an intent detection model with dual encoders that are pre-trained on 654 million (input, response) pairs from Reddit.

- **CONVEBERT** (Mehri et al., 2020): a BERT-base model which has been trained on a large open-domain dialogue corpus. CONVEBERT achieved improvements over a vanilla BERT architecture and state-of-the-art results on a few task-oriented dialogue tasks.

- **CONVEBERT+Combined** (Mehri et al., 2021): a CONVEBERT-base model trained to improve similarity matching of training examples, i.e., Match-QQ. Additionally, the model trains with observers for transformer attention and conducts task-adaptive self-supervised learning with mask language modeling (MLM) on the intent detection datasets. **Combined** represents the best MLM+Example+Observers setting in the referenced paper.

- **DNNC** (Zhang et al., 2020): Discriminative Nearest Neighbor Classification (DNNC) model is trained to find the best-matched example from the training set through similarity matching (Match-QQ). The model conducts data augmentation during training and boosts performance by pre-training on three natural language inference tasks.

- **CPFT** (Zhang et al., 2021b): Contrastive Pre-training and Fine-Tuning (CPFT) is a two-stage intent-detection architecture. During the first stage, the model learns with a self-supervised contrastive loss on a large set of unlabeled queries. In the second stage, the model learns with supervised contrastive loss to pull together query representation from the same intent (Match-QQ). The inference is done via a classification head that is added and trained during the second stage.

## 4 RESULTS

Table 2 lists the results on the three datasets described in Section 3.1. QAID with RoBERTa-base achieved the best results across all datasets and shots. Notably, increasing the model size from RoBERTa-base to RoBERTa-large resulted in additional significant improvement across all datasets. For 5-shot, QAID with RoBERTa-base improves over CPFT, which achieved the best results reported so far, by more than 4 points on the BANKING77 dataset which is translated to $30.64\%$ in error rate reduction (ERR). Similarly, QAID achieves ERR of $8.5\%$ and $18.9\%$ over CPFT for the CLINC150 and HWU64 respectively. We attribute our improvement to three key differences between our method and CPFT. (1) Our problem formulation kept the class representation the same during training and inference. In other words, we didn't train a classification layer for inference. (2) Incorporating answers as data points contributes an additional discriminating signal. (3) The token-level late interaction score is shown to perform better as our ablation experiments demonstrate in Section 4.1. Moreover, our standard deviations (std) are consistently lower than those of DNNC and CPFT with the highest std of 0.15 and average std of 0.07. We believe the reason for

| Model | CLINC150 | | BANKING77 | | HWU64 | |
|---|---|---|---|---|---|---|
| | 5 | 10 | 5 | 10 | 5 | 10 |
| Classifier (RoBERTa-base) | 87.68 | 91.22 | 74.46 | 83.79 | 73.52 | 82.62 |
| ColBERT | 82.03 | 88.10 | 72.71 | 79.25 | 74.98 | 81.78 |
| USE+CONVERT (Casanueva et al., 2020) | 90.49 | 93.26 | 77.75 | 85.19 | 80.01 | 85.83 |
| CONVBERT (Mehri et al., 2020) | - | 92.10 | - | 83.63 | - | 83.77 |
| CONVBERT + Combined (Mehri et al., 2021) | - | 93.97 | - | 85.95 | - | 86.28 |
| DNNC (Zhang et al., 2020) | 91.02 | 93.76 | 80.40 | 86.71 | 80.46 | 84.72 |
| CPFT (Zhang et al., 2021b) | 92.34 | 94.18 | 80.86 | 87.20 | 82.03 | 87.13 |
| QAID (RoBERTa-base) | **93.41** | **94.64** | **85.25** | **88.83** | **85.52** | **87.98** |
| QAID (RoBERTa-large) | 94.95 | 95.71 | 87.30 | 89.41 | 87.82 | 90.42 |

Table 2: Accuracy results on three ID datasets in 5-shot and 10-shot settings. Baseline results are from the original papers except for RoBERTa Classifier and ColBERT.

the low std in the results is the combination of batch contrastive loss with data augmentation and the fine-grained late-interaction similarity score. Accordingly, our improvements are significant according to an unpaired t-test with a p-value of $1e^{-4}$. An additional advantage of our method is its efficiency. QAID pre-training run-time takes about two hours and it has to run only once for all of our targets. Our fine-tuning takes only ten minutes on one NVIDIA V100 GPU, compared to three hours of fine-tuning of DNNC. Another important aspect of our results is the effect of scaling from RoBERTa-base to RoBERTa-large, which resulted in significant improvements in both 5 and 10-shot scenarios across all datasets, aligned with results showing larger models generalize better from small data (Bandel et al., 2022). Moreover, in some cases scaling the model was more beneficial than additional examples. Namely, RoBERTa-large in 5-shot surpasses RoBERTa-base in 10-shot.

## 4.1 ABLATION TESTS

In this section, we describe several ablation studies demonstrating the importance of our method components and the main factors that contribute to our improvement over ColBERT.

We present our ablation results in Table 3. We start by analyzing the improving effect of the pre-training (PT) and batch contrastive training on ColBERT. We can see that both stages boost the performances considerably across all settings. It is noticeable that the **pre-training** (row **ColBERT with PT**) improves more in the 5-shot than in the 10-shot setting with deltas of 4.59 and 2.00 points on average, respectively. This result is consistent with the observation that a model with better query representation is essential when only a few examples are available (Choshen et al., 2022). **Batch contrastive training** (row **ColBERT with batch contrastive**) improves performance in most settings, with an average improvement of 5.74 points over ColBERT. We attribute this improvement to two major enhancements that batch contrastive training introduces. The first is the shift from a model that learns only Match-QA to a model that learns both Match-QA and Match-QQ. The second is the improved technique of batch contrastive loss over triplet loss that allow to process many positive and negative examples at once and intrinsic ability to perform hard positive/negative mining (Khosla et al., 2020). We note that this change has a minor effect on the training time as it relies on representations calculated in the batch and has no effect on the inference time.

In addition, we study some modifications of QAID to understand their effect. To further investigate the effect of **batch contrastive** we train QAID with N-pairs loss (row **QAID - N-pairs loss**) with in-batch negatives, a widely used loss in retrieval models, e.g. DPR and RocketQA (Karpukhin et al., 2020; Qu et al., 2021). In this setting, each query has one positive example, which in our case is the intent name, and multiple irrelevant (negative) examples, either different queries or intent names. This method differs from our supervised batch contrastive loss which allows many positive examples. Our results show that replacing QAID supervised batch contrastive loss with N-pairs loss leads to a decrease of more than 1 point on average. These results support our claim that retrieval models can benefit from adopting supervised batch contrastive loss. When we conducted fine-tuning training with and without **auxiliary tasks** (MLM and classification losses), we found

that the auxiliary tasks increased QAID accuracy by 0.43 on average. Interestingly, the increase was more pronounced as the dataset contained fewer domains, 0.72, 0.48, and 0.10 of average improvement on Banking77, Clinc150, and HWU64, respectively. We also examine the performance of our method without **pre-training**, row **QAID w/o PT**. Results indicate that our method achieves an average improvement of 3.53 points compared to CPFT without pre-training (**CPFT w/o PT**) in Table 3. This result emphasizes the superiority of our proposed method as a fine-tuning method. Additionally, to better understand the role of the **data augmentation** module in our training, we conduct an experiment where we did not apply the data augmentation module. Results, **QAID w/o data augmentation**, show that by augmenting the data we improve the results by about a third of a point on average. In the 5-shot setting, the improvement is about 0.75 points on average, and in the 10-shot setting, the effect is inconsistent. Those results can indicate that data augmentation is more beneficial where less data is available.

We experiment with replacing the **similarity score** in QAID with the cosine similarity of the CLS token (row **QAID - Cosine Similarity**) instead of the token-level late interaction (Khattab & Zaharia, 2020) (row **QAID**). We can see that using the token-level late interaction achieves higher results across all datasets and shots. We ascribe this improvement to the fine-grained nature of the late-interaction score that enables detailed token-level comparison. This score presents an efficient alternative to the costly cross-attention scoring that most Match-QQ methods use.

Finally, we discuss our **inference method**. We experiment with indexing and predicting based on both queries and answers, i.e., using Match-QQ and Match-QA in the inference stage as we do in training. Inference based only on Match-QA achieve slightly better (0.07) results on average, with an average improvement of 0.18 and 0.17 on Banking77 and Clinc150, respectively, and an average decrease of 0.14 on HWU64. These results indicate that our training method achieves answer representation that reflects the distribution of both training queries and answers. Therefore, allowing a more efficient inference that relies only on the answers' representations.

| Model | CLINC150 | | BANKING77 | | HWU64 | |
|---|---|---|---|---|---|---|
| | 5 | 10 | 5 | 10 | 5 | 10 |
| ColBERT | 82.03 | 88.10 | 72.71 | 79.25 | 74.98 | 81.78 |
| ColBERT with PT | 89.31 | 90.85 | 75.73 | 80.42 | 81.20 | 84.88 |
| ColBERT with batch contrastive | 89.92 | 93.17 | 81.41 | 86.68 | 80.36 | 85.54 |
| CPFT w/o PT (Zhang et al., 2021b) | 88.19 | 91.55 | 76.75 | 84.83 | 76.02 | 82.96 |
| QAID - N-pairs loss | 92.38 | 93.70 | 84.47 | 87.25 | 84.19 | 87.13 |
| QAID w/o PT | 90.52 | 93.26 | 81.61 | 86.20 | 80.45 | 86.48 |
| QAID w/o data augmentation | 93.21 | **94.64** | 84.34 | 88.46 | 84.37 | **88.46** |
| QAID - cosine similarity | 92.73 | 93.71 | 84.11 | 87.24 | 84.04 | 87.70 |
| QAID | **93.41** | **94.64** | **85.25** | **88.83** | **85.52** | 87.98 |

Table 3: Accuracy results of our ablation experiments.

## 5 RELATED WORK

### 5.1 FEW-SHOT INTENT DETECTION

Task adaptive pre-training is a common strategy to face the data scarcity problem in few-shot intent detection classification. Predominant approach for task adaptive pre-training models leverages self-supervised mask language modeling training on large dialogues datasets (a few hundred million dialogues) and on the domain itself to tackle few-shot intent detection (Casanueva et al., 2020; Mehri et al., 2020; 2021). Shnarch et al. (2022) showed that unsupervised clustering helps better than MLM pre-training. DNNC (Zhang et al., 2020) pre-trains their system on annotated pairs from natural language inference (NLI) leveraging BERT (Devlin et al., 2018) pairwise encoding. Then they model intent detection as a kNN problem with $k = 1$ where the pre-trained NLI model learns to predict the similarity score for pair of queries. However, this model is computationally expensive as it fully utilized training examples in both training and inference.

The CPFT work, suggested by Zhang et al. (2021b), shares some similarities with our approach. when the two main ones are: the two-stage training process and the use of batch contrastive loss. However, we have several key differences where we extend upon their method. Those differences make our method more effective and efficient, especially when the task involves a large number of intents. Firstly, we reformulate the few-shot ID classification task as a retrieval task. To that end, we adopt an efficient Dual-Encoder-based retrieval architecture - ColBERT (Khattab & Zaharia, 2020), and a late-interaction similarity score. Moreover, we treat the intent names as the answers we wish to retrieve. Secondly, we adjust the training method to learn both Match-QQ and Match-QA similarities. Finally, we adopt a retrieval-based inference based on the similarity between the incoming query and the intent names, therefore we are not required to train an additional classification head.

Zhang et al. (2022b) design two stage training with batch contrastive loss and add explicit regularization loss directing the feature space towards isotropy. They report high 5-way few-shot results on the same benchmarks we use. Nevertheless, when evaluating their model accuracy the results are much lower than ours (about $16\%$ and $9\%$ lower on Banking77 and HWU64 respectively).

## 5.2 Intent name

The idea of exploiting class names was proposed in the setting of zero and few-shot classification by a few past works (Meng et al., 2020; Yin et al., 2019; Basile et al., 2021). Yin et al. (2019) propose to formulate text classification tasks as a textual entailment problem (Dagan et al., 2005). This mapping enables using a model trained on natural language inference (NLI) as a zero-shot text classifier for a wide variety of unseen downstream tasks (Gera et al., 2022). Zhong et al. (2021) map the classification tasks to a question-answering format, where each class is formulated as a question and given as a prompt, and the decoder probabilities of the "Yes" and "No" tokens correspond to a positive or negative prediction of the class.

In our work, we cast the classification problem to the task of question answering retrieval and treat a much larger number of classes than these works tackle, which is usually up to twenty.

## 5.3 Batch Contrastive Learning

Batch contrastive training was shown to achieve improved representation and perform better than contrastive losses such as triplet, max-margin, and the N-pairs loss (Khosla et al., 2020). Gunel et al. (2021); Gao et al. (2021b) suggest incorporating batch contrastive learning to train the encoder in natural language processing tasks. Gao et al. (2021b) designed a simple contrastive learning framework through dropout augmentation. They trained on NLI data to achieve state-of-the-art results on unsupervised and full-shot supervised semantic textual similarity (STS) tasks (Agirre et al., 2012; 2015; 2016). Liu et al. (2021a) suggest MirrorBERT, a self-supervised framework with two types of random data augmentation: randomly erase or mask parts of the texts during tokenization, and representation-level augmentation through built-in encoder dropout. We differ from those works as we target the few-shot intent detection task. Moreover, we adjust the late interaction score from Khattab & Zaharia (2020) to achieve cross-attention-like similarity scores. We also showed that class names can serve as an additional augmentation that can be the base for inference prediction.

## 6 Conclusions

In this paper, we present QAID, Question Answering inspired Intent Detection system, that models the few-shot ID classification as a question-answering retrieval task, where utterances serve as questions and intent names as answers. We train QAID with a two-stage training schema with batch contrastive loss. Results show that replacing ColBERT triplet loss with batch contrastive loss leads to a considerable improvement. We assert that a contributing factor to this effect is the shift to learning Match-QQ and Match-QA representations. We leave for further research to investigate this effect on retrieval tasks. Moreover, our results show that incorporating token-level similarity scores in contrastive loss outperforms the common cosine similarity score without a notable increase in training time. We encourage future research to utilize this type of contrastive loss in other tasks and investigate its effect. Finally, our results on three few-shot ID benchmarks show that QAID achieves state-of-the-art performance.

ACKNOWLEDGEMENTS

We thank Leshem Choshen and Ariel Gera for their helpful feedback as we pursued this research.

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
