# OpenReview forum: "QAID: Question Answering Inspired Few-shot Intent Detection"
_ICLR.cc/2023/Conference — ICLR 2023 poster_

### Official Review · Reviewer_ZHT4 · 2022-10-24

**Confidence:** 4
**Correctness:** 3
**Technical Novelty And Significance:** 2
**Empirical Novelty And Significance:** 2
**Recommendation:** 6

**Clarity, Quality, Novelty And Reproducibility:**

The paper is well written and the results are good. This work is for balanced few-shot datasets. It will be interesting to see if the proposed techniques work on imbalanced datasets as well.


**Strength And Weaknesses:**

Strengths

S1: It is an interesting idea to treat ID problem as a question-answering task.

S2: Utilization of Token level domain knowledge in a self-supervised fashion along with queries adaptation to dialogue domain is interesting.

S3: Masked language modeling has also been employed as auxiliary loss.

S4: Intent names have been employed for generating extra supervision signals.

S5: There are detailed experiments in this work, with an interesting insight that ColBERT triplet loss with batch contrastive loss leads to a considerable improvement.

S6: The paper is well written and easy to follow.


Weaknesses
W1: Why a balanced dataset is the requirement for this work. If the dataset is imbalanced, how will the proposed method behave?

W2: Typos: In abstract: “self supervise manner.”


**Summary Of The Paper:**

In this work, the authors reformulate the problem of intent detection as a question-answering task by treating utterances and labels as questions and answers. A two stage training schema is employed by utilizing question-answering retrieval architecture and batch contrastive loss. The first of the training stages is for learning better query representations, while the second stage is for optimizing the contextualized token-level similarity scores between queries and answers from the same intent. State-of-the-art results are reported on three few-shot intent detection benchmarks.


**Summary Of The Review:**

Overall, this is an interesting paper and addresses the challenging intent detection task in a question-answering setting.

---

> ### Author Response · Authors · 2022-11-17
> **Imbalanced data experiment**
>
> Thank you very much for your helpful review. We are glad you find our paper to be interesting, well-written, and easy to follow, and recognize our innovation in adopting question-answering for the challenging ID classification task together with our use of Intent names for extra supervision signals. We are happy you think that our experiments are detailed, interesting, and insightful, especially the result that ColBERT can be improved by utilizing supervised contrastive loss.
>
> "Why a balanced dataset is the requirement for this work. If the dataset is imbalanced, how will the proposed method behave?"
>
> Answer: In our paper, we examine the common scenario where we learn an Intent Detection model with only K examples per intent as we explained at the beginning of section 2. This allows for a comparison across different models and approaches, especially to the previous SOTA results. Nevertheless, although the scenario of imbalanced data is out of this work scope, it is a realistic and interesting scenario that is worth checking. For that experiment, we randomly sample 50% of the 10-shot training data, creating a new synthetic (intentionally) imbalanced training data (the distribution for BANKING77 appears below). Then we use it to train a regular RoBERTa classifier as well as our QAID method. Results appear in the Table below. We can see that QAID outperforms RoBERTa classifier by close to 8 points on average.
>
> | Model / DATA | CLINC150 | BANKING77 | HWU64 |
> | --- | --- | --- | --- |
> | synthetic imbalance data | 50% of 10 | 50% of 10 | 50% of 10 |
> | Classifier (RoBERTa-base) | 76.41 | 60.36 | 56.75 |
> | QAID | 81.42 | 70.60 | 65.16 |
>
>
> Distribution for the synthetic imbalance BANKING77 data:
>
> | Number of examples per intent | 1 | 2 | 3 | 4 | 5 | 6 | 7 | 8 | 9 | 10 |
> | --- | --- | --- | --- | --- | --- | --- | --- | --- | --- | --- |
> | Intents with that # of examples | 8 | 13 | 8 | 8 | 8 | 6 | 5 | 9 | 8 | 4 |

---

### Official Review · Reviewer_73BF · 2022-10-24

**Confidence:** 3
**Clarity, Quality, Novelty And Reproducibility:** 1. Where are the ablation results for…
**Correctness:** 4
**Technical Novelty And Significance:** 2
**Empirical Novelty And Significance:** 3
**Recommendation:** 6

**Strength And Weaknesses:**

Strengths:

1. The paper reports SOTA results in three benchmark intent identification tasks for both 5-shot and 10-shot setting.

2. The paper shows ablations to support the claims on supervised batch contrastive training and late interaction scores.

3. The paper is well written and easy to understand


Weakness:

1. The ablations can be improved. The paper can show ablation to understand the component of each component and the impact it has on the intent id task. The missing ablations are: QAID – batch contrastive,  QAID – intent names (not sure if this is possible) and QAID – data augmentation. But it would be good to show results by removing one component at a time.

**Summary Of The Paper:**

This paper presents a Question-Answering inspired Intent Detection System (QAID). The paper treats the intent identification task as a Question-Answering retrieval task by treating the utterances and the intent names as queries and answers respectively. QAID adapts ColBERT architecture from prior work and replaces the loss function with batch contrastive loss. The novelty of this work is in converting intent id task to Question-Answering form and using the intent names as answer, and using supervised batch contrastive training for finetuning LM on this task.

**Summary Of The Review:**

Overall, this paper builds on top of prior work (especially ColBERT and CPFT) but uses intent names for retrieval. The paper shows the benefit of combining these techniques and presents SOTA results and shows good ablations.

---

> ### Author Response · Authors · 2022-11-17
> **Proposed Ablations: 1) QAID w/o batch contrastive 2) QAID w/o intent names 3) QAID w/o data augmentation**
>
> Thank you very much for your helpful review. We are glad you find our paper to be well-written and easy to understand, recognize that our method achieves SOTA results on Intent Detection benchmarks, and find our ablations to support the claims on supervised batch contrastive training and late interaction scores.
>
> "Where are the ablation results for "data augmentation". The paper says that data augmentation results are inconsistent but the results are not shown in Table 3."
>
> Answer: First, to clarify, the ablation results for "data augmentation", meaning ColBERT w/ data augmentation, indeed, were not presented in the paper. We add those results in the Table below. Those results show an inconsistent effect, improved results in some cases, and lesser results in other cases. The QAID w/o data augmentation ablation can further explain the importance of the data augmentation component in our supervised batch contrastive training.
>
> "The ablations can be improved. The paper can show ablation to understand the component of each component and the impact it has on the intent id task. The missing ablations are: QAID w/o batch contrastive, QAID w/o intent names (not sure if this is possible) and QAID w/o data augmentation. But it would be good to show results by removing one component at a time."
>
> Answer: Regarding the proposed ablation study, we didn't conduct QAID w/o batch contrastive, QAID w/o intent names, in the paper because there are no simple controlled experiments that allow for those modifications, as explained below for each ablation. Nevertheless, here we tried to come up with similar variants that can demonstrate the importance of the relevant components.
>
> QAID w/o batch contrastive: To move from batch contrastive loss to triplet loss we need to construct the triplets before training but our data augmentation happened dynamically, so this modification is not straightforward. Instead, we conduct ablation where we use N-pairs loss with "in batch negatives". In that setting, each query has one answer as a positive example and many negative examples that come from different instances within the batch. The results of that experiment show that replacing QAID supervised batch contrastive loss with N-pairs loss causes a decrease of more than 1 point on average. The new results appear in the updated version of the paper and the Table below. These results further demonstrate the importance of the supervised batch contrastive loss.
>
> QAID w/o intent names: We use the intent names for the inference, so removing the intent names will demand a modification that will prevent a controlled experiment. To overcome this, we tried two different approaches: 1) Use the intent names only for inference. 2) Generate new intent names based on the queries. 1) This experiment shows results that are 10 points lower than QAID. Those results show that without training on the intent names there is a discrepancy between the training and inference, but they don't demonstrate the importance of intent names in QAID. 2) We use BERT-key to generate intent names based on the queries. We then apply QAID on the original queries with the generated intent names. The results are shown in the Table below. We can see a decrease of more than 5 points, indicating the importance of quality intent names.
>
> QAID w/o data augmentation: In this ablation, we remove the data augmentation module, by using only query with its intent names as augmentation. This ablation results show that using data augmentation improves results by about 0.35 points on average, improving about 0.75 points on average in the 5-shot setting and appearing to have an inconsistent effect in the 10-shot setting. Those results appear in the updated version of the paper as well as in the Table below. From a broader perspective, we can add that while the data augmentation may have moderately contributed to the supervised stage, it played a major role in the self-supervised stage. This contribution is evident from the inferior results without the pre-training stage.

---

> ### Author Response · Authors · 2022-11-17
> **Ablation study results**
>
> |       Model / DATA                  |     CLINC150  |          |     BANKING77  |          |     HWU64  |          |
> |-------------------------------------|:-------------:|:--------:|:--------------:|:--------:|:----------:|:--------:|
> |                                     |   5           |   10     |   5            |   10     |   5        |   10     |
> |     ColBERT                         |   82.03       |   88.10  |   72.21        |   79.25  |   74.98    |   81.78  |
> |     ColBERT w/ data augmentation    |   82.09       |   87.14  |   72.58        |   79.69  |   75.64    |   81.78  |
> |     QAID w/ N-pairs loss            |   92.38       |   93.70  |   84.47        |   87.25  |   84.19    |   87.13  |
> |     QAID w/ ans. In inference only  |   85.14       |   87.10  |   78.00        |   81.41  |   71.38    |   75.63  |
> |     QAID w/ BERT-key intent names   |   87.53       |   92.23  |   78.62        |   85.51  |   74.49    |   84.32  |
> |     QAID w/o data augmentation      |   93.21       |   94.64  |   84.34        |   88.46  |   84.37    |   88.46  |
> |     QAID                            |   93.41       |   94.64  |   85.25        |   88.83  |   85.52    |   87.98  |

---

### Official Review · Reviewer_rUR4 · 2022-11-05

**Confidence:** 3
**Correctness:** 3
**Technical Novelty And Significance:** 2
**Empirical Novelty And Significance:** 2
**Recommendation:** 5

**Clarity, Quality, Novelty And Reproducibility:**

The writing in the paper can be improved with a better description of the two-stage contrastive training subsection. (Some terminologies in there, such as anchor/pivot, are hard to understand). A more comprehensive schematic illustration (compared to Figure 1) will help. Further, adding some qualitative examples showing why the proposed changes show improvements can help strengthen the motivation.

The main originality in the paper is the reformulation intent detection as a retrieval task. Other proposed contributions are mainly about combining ideas from retrieval literature with those recently proposed for few-shot intent detection in the CPFT paper.


**Strength And Weaknesses:**

Strengths:

Formulating as a retrieval task helps make use of current SOTA techniques in retrieval such as late interaction scores and batch contrastive loss. The idea of using a retrieval-based dual encoder for this task is interesting from an efficiency perspective too, since the intent names (which are the answers) can be pre-computed during inference. This overcomes the high latency of cross-attention based approaches for which the processing time scales with the number of intents.

Weaknesses:

1) It's not entirely clear why augmentation is needed and what is the motivation for / benefit from using the data augmentation module

2) Since CPFT seems to be most similar to this paper, the authors need to highlight better what the main differences are with CPFT. From reading the paper, the impression I get is that the authors just incorporate the few-shot intent detection methodologies used in CPFT (such as self-supervised contrastive pre-training and fine-tuning) into a dual-encoder based retrieval formulation (by using ColBERT)

3) It is not surprising to see benefits from using late-interaction scores and batch contrastive training (which is one of proposed contributions). ColBERT/ColBERT-V2, which are based on late-interaction scores, have already been shown to outperform cosine similarity based matching in dual encoder models. Further, batch contrastive loss is already a popularly used training technique for dual-encoder based retrieval models (such as DPR, RocketQA)

**Summary Of The Paper:**

The paper reformulates intent detection as a retrieval task, by treating utterances and intent names as questions and answers respectively. The authors leverage a dual-encoder based retrieval architecture for few-shot intent detection, with late-interaction scores (as used in ColBERT) and batch contrastive training.

**Summary Of The Review:**

Overall the paper shows good (and statistically significant) improvements over existing methods for few-shot intent detection. However, the novelty in the paper is limited as it mainly involves combining existing techniques in a retrieval-based formulation (such as batch contrastive loss and late-interaction scores) with methodologies for few-shot intent detection proposed in CPFT (such as self-supervised contrastive pre-training and fine-tuning)

---

> ### Author Response · Authors · 2022-11-17
> **3) Contribution in late-interaction scores and batch contrastive training**
>
> 3. "It is not surprising to see benefits from using late-interaction scores and batch contrastive training (which is one of proposed contributions). A) ColBERT/ColBERT-V2, which are based on late-interaction scores, have already been shown to outperform cosine similarity-based matching in dual encoder models. B) Further, batch contrastive loss is already a popularly used training technique for dual-encoder based retrieval models (such as DPR, RocketQA)"
>
> Answer: A) Late Interaction: We agree that late interaction proved efficient and effective in previous work. To the best of our knowledge, we are the first to conduct a controlled experiment with a full-size model where the similarity metric is the only parameter that was changed. We note that ColBERT paper showed an ablation study with a comparison of ColBERT with late interaction and BERT, both with five layers. Although this experiment can hint at the effectiveness of late interaction score, the five layers setting is more suited to word representation than sentence representation and thus can favor late interaction over [CLS] representations cosine similarity score, but can't necessarily support the same conclusion in the regular BERT setting.
>
> Moreover, as far as we know, we are the first to apply late interaction-based learning to a classification problem which is different from the task of passage retrieval in which late interaction was used. Additionally, we measure the similarity between queries and intent names that are considered very different. While the queries are naturally structured, the intents are human-annotated keywords. For those reasons, we believe that our results are new and expand upon known previous results.
>
> B) Batch Contrastive: The mentioned methods, DPR and RocketQA, indeed apply a similar yet different approach, named N-pairs loss. In that setting, each query has one positive example, which in our case is the intent name, and multiple irrelevant (negative) examples, either different queries or intent names. This method differs from our supervised batch contrastive loss which allows many positive examples. Additionally, those methods used "in-batch negatives", meaning, for each query in the batch, the positive example of all of the different questions in the batch act as negative examples of the current question. In our method, for each query, we use all of the batch instances with the same label as positive examples.
>
> Another relevant point regarding supervised batch contrastive training is the way we apply it. In our method, we learn Match-QQ and Match-QA similarities, instead of only Match-QA as used in those methods.
>
> To better understand the difference between our loss function and N-pairs loss function we add a new ablation study that uses N-pairs loss with "in-batch negatives". We show that replacing QAID supervised batch contrastive loss with N-pairs loss causes a decrease of more than 1 point on average. The new results appear in the updated version of the paper and the Table below. These results support our claim that retrieval models can benefit from adopting supervised batch contrastive loss as R4 (ZHT4) pointed out.
>
> |       Model               |     CLINC150  |          |     BANKING77  |          |     HWU64  |          |
> |---------------------------|:-------------:|:--------:|:--------------:|:--------:|:----------:|:--------:|
> |                           |   5           |   10     |   5            |   10     |   5        |   10     |
> |     QAID                  |   93.41       |   94.64  |   85.25        |   88.83  |   85.52    |   87.98  |
> |     QAID w/ N-pairs loss  |   92.38       |   93.70  |   84.47        |   87.25  |   84.19    |   87.13  |
> |     QAID w/o data aug.    |   93.21       |   94.64  |   84.34        |   88.46  |   84.37    |   88.46  |
>
> Chen, T., Kornblith, S., Norouzi, M., & Hinton, G. (2020, November). A simple framework for contrastive learning of visual representations. In International conference on machine learning (pp. 1597-1607). PMLR.‏
>
> Prannay Khosla, Piotr Teterwak, Chen Wang, Aaron Sarna, Yonglong Tian, Phillip Isola, Aaron Maschinot, Ce Liu, and Dilip Krishnan. Supervised contrastive learning. Advances in Neural Information Processing Systems, 33:18661–18673, 2020.
>
> Beliz Gunel, Jingfei Du, Alexis Conneau, and Ves Stoyanov. 2020. Supervised contrastive learning for pretrained language model fine-tuning. arXiv preprint arXiv:2011.01403.
>
> Tianyu Gao, Xingcheng Yao, and Danqi Chen. 2021. Simcse: Simple contrastive learning of sentence embeddings. EMNLP.

---

> ### Author Response · Authors · 2022-11-17
> **1) Benefit of data augmentation 2) Differences from CPFT**
>
> Thank you for your helpful review. We are glad you found our adoption of retrieval architectures for the task of intent detection interesting and helpful from an efficiency perspective. Indeed, one of the core contributions of our paper, is leveraging retrieval-based dual encoders combined with late interaction scores, to overcome the high latency of cross-attention-based approaches in the context of text classification tasks such as intent detection.
>
> Regarding the points you have raised:
>
> 1) "It's not entirely clear why augmentation is needed and what is the motivation for / benefit from using the data augmentation module"
>
> Answer: The data augmentation module is a common and essential component for self-supervised contrastive training (Chen et al.). In this scenario, no labels are available, and augmenting the training examples allows us to create positive variations of the training examples to train on. Accordingly, for the self-supervised pre-training, we use the data augmentation module to train the encoder toward better queries representations in the dialog domain.
>
> In the scenario of supervised batch contrastive training, we face a few-shot classification task where data scarcity is the main difficulty. In this context, adding data augmentation provides an additional signal to the model to learn a more robust query representation. This method was used in a few previous papers, in both vision (Khosla et al., 2020) and language (Gunel et al., 2021; Gao et al., 2021b). Moreover, to better understand the role of the data augmentation module in our training, we conduct a new experiment where we did not apply the data augmentation module. Those results, appearing in the current version of the paper and the table below, show that by augmenting the data we improve the results by about a third of a point on average. In the 5-shot setting, the improvement is about 0.75 points on average, and in the 10-shot setting, the effect is inconsistent.
>
> 2) "Since CPFT seems to be most similar to this paper, the authors need to highlight better what the main differences are with CPFT. From reading the paper, the impression I get is that the authors just incorporate the few-shot intent detection methodologies used in CPFT (such as self-supervised contrastive pre-training and fine-tuning) into a dual-encoder based retrieval formulation (by using ColBERT)"
>
> Answer: Thank you for this comment. Indeed, we share some similarities with CPFT, the two main ones are; two-stage training and batch contrastive loss. But we also have several key differences where we extend upon their method. Those differences make our method more effective and efficient, especially in cases with a large number of intents. To allow for a more efficient model that can handle many intents, we used a dual-encoder-based retrieval architecture that can cope with many passages/answers. To adopt this architecture, we use the intent names as answers. Accordingly, we needed to adjust the training method to learn both Match-QQ and Match-QA similarities. Eventually, we used retrieval-based inference that is based on the similarity between the incoming query and the intent names. In the results section, we showed that our model achieves better performances than CPFT with and without the pre-training stage.
>
> Those adaptations are not trivial and present a few new concepts that weren't shown in previous works as far as we know. 1) Dual-Encoder-based retrieval models can be adapted to classification tasks and show effective and efficient performances. 2) Class labels can be treated as answers in this setting and provide a helpful signal. 3) Learning Match-QQ and Match-QA-based representation can be helpful and surpass only Match-QA signals. 4) Classification with batch contrastive loss can use retrieval-based inference instead of using classification head that demands additional training stage. We clarify those points and highlight the differences with CPFT in the current version.

---

### Official Review · Reviewer_uX1q · 2022-11-07

**Confidence:** 3
**Correctness:** 3
**Technical Novelty And Significance:** 2
**Empirical Novelty And Significance:** 3
**Recommendation:** 6

**Clarity, Quality, Novelty And Reproducibility:**

Clarity: The paper is clearly written.
Quality: The statements are well supported. The empirical results are solid.
Novelty: The paper adapts the method from question answering to intent detection. The adaptation itself is novel although the proposed methods are a combination of existing techniques.
Reproducibility: It requires some effort to reproduce the results.

**Strength And Weaknesses:**

Strength:

- The paper uses and adapts techniques from QA and answer retrieval to the task of ID. The method is effective and efficient on three benchmark datasets when compared with various baselines.
- The detailed ablation study verifies the necessity of stages and component of proposed framework.
- The paper is well-written and easy to follow.

Weakness:
- Given the number of intents being only a few hundreds at most, is Faiss really necessary for inference?
- n has two meanings in section 2.1: one for batch size and another for number of tokens.

**Summary Of The Paper:**

The paper proposes QAID, Question Answering inspired Intent Detection system, which models the intention detection classification as a question-answering task. The model uses two stages of training: a pretraining for better query representation and finetuning on few-shot labels of query and answers (name of intents). Certain choices of model, such as token-level similarity, batched contrastive learning, and inference with answers only, are verified by detailed ablation studies. The proposed method achieves SOTA  performance on three intention detection dataset from DialoGLUE.

**Summary Of The Review:**

The paper is inspired by the development of ColBERT in QA task and propose a variant of the model for the task of intent detection. The proposed model makes several adaptations including batch contrastive loss and  signal from the intent names. The proposed method achieves SOTA performance on three few-shot ID benchmarks. And the ablation study proves the necessity of various components in the system.

---

> ### Author Response · Authors · 2022-11-17
> **Efficiency condensing**
>
> Thank you very much for your helpful review. We are glad you find our paper well-written and easy to follow, our method effective and efficient, and verified by a detailed ablation study.
>
> Regarding the questions you have raised:
>
> 1) "Given the number of intents being only a few hundred at most, is Faiss really necessary for inference?"
>
> Answer: Intent Detection models are widely used in commercial systems, making their efficiency critical. Accordingly, we chose to use Faiss to allow users to scale the number of intents while maintaining efficiency, as rightfully pointed out by R2 (rUR4). In our scenario, you are right. We have only a few hundred intents, thus a few thousand token representation vectors that Faiss stores, making the efficiency improvement quite low compared to cases with thousands of intents.
>
> 2) "n has two meanings in section 2.1: one for batch size and another for number of tokens."
>
> Answer: thank you for this comment, we will fix our notation in the next version.

---

### Decision · Program_Chairs · 2023-01-20

**Decision:**

Accept: poster

**Justification For Why Not Higher Score:**

The primary concern is that this work is very specific to intent classification, which is an important problem, but not necessarily of wide interest. In principle, this work could be expanded to other settings where scale is important and labels have meaning -- which would likely increase the potential impact of the work.

**Justification For Why Not Lower Score:**

Intent classification is a NLP problem of significant interest and potential impact -- and the solution is easy to understand and performs well. Additionally, the ablation studies do a good job at supporting the claims of this paper.

**Metareview: Summary, Strengths And Weaknesses:**

The authors propose "Question-Answering inspired Intent Detection" (QAID), which frames (few-shot) intent detection as a QA pair of [utterance=question] and [answer=intent names] and builds on recent methodological advances from the QA community to achieve state-of-the-art few-shot intent classification results on widely-used benchmark datasets. Specifically, they adapt the ColBERT architecture by replacing triplet loss with batch contrastive loss, first augmenting the data with widely-used corruption techniques (MLM and dropping representation/inner-neuron elements) and using this data to perform LM fine-tuning and supervised training on the intent names -- noting that the encoder is used on the intent names during supervised training to take advantage of 'dataless' classification advantages. Since this is a QA inspired system, they also use the common approach of a dual encoder with 'late interaction scores' via FAISS-based retrieval. Using the DialoGLUE intent detection data (CLINC150, BANKING77, HWU64) and several recent baseline systems, they show that they establish a new SoTA on these datasets. Secondly, they perform ablation studies to show the relative contribution associated with each subcomponent.

The consensus strengths of this work include:
- QAID builds on recent notable improvements to build a few-shot intent detection algorithm, proposing non-trivial modifications specific to aspects of intent detection (e.g., intent name, suitable data augmentation to support contrastive learning).
- The primary experiments exhibit SotA few-shot intent detection on the widely-used benchmark datasets for this task and the ablation studies support the claimed improvements in the paper.
- The paper is well-written, well-contextualized, and easy to follow (especially when reading some fo the background work).
- The proposed solution is methodological in nature, but ostensibly would easily scale to commercial solutions if desired.

The weaknesses discussed in the reviews (and my own reading) of this work include:
- Since this work relies heavily on recent QA work and conceptually uses methods similar to CPFT for training, there were some concerns regarding novelty. I believe this was addressed in the rebuttal and encourage the authors to include these in the final revision.
- There were some detailed technical questions in the reviews -- that were sufficiently answered in the rebuttal.
- The modifications are somewhat specific to intent classification (or at least to 'dataless classification' settings where the class label can be projected into a suitable embedding space). Thus, it isn't clear that this will 'back-translate' to improve QA nor inspire similar works. This isn't a criticism as it isn't the point of the paper, but does indicate limited scope -- and that it may be more suitable for a NLP-specific venue as opposed to a more ML venue.

Overall, the consensus is that this is a work that is 'correctly inspired' by recent QA methods adapted using recent intent classification works to achieve SotA results that are likely scalable all the way to industry settings. Overall, this is a elegant, easily understood approach and the paper is well-executed. As an intent detection work, this will very likely be a notable baseline regarding future work. Thus, we lean toward acceptance. That being said, the results are likely unique to intent classification and might be of lesser interest to the broader ML community.


**Note From Pc:**

if the above contains the word "oral" or "spotlight" please see: "oral" presentation means -> notable-top-5% and "spotlight" means -> notable-top-25%. As stated in our emails, we are disassociating presentation type from AC recommendations

**Summary Of Ac-Reviewer Meeting:**

The concern raised during discussion is reflected in the reviews. Specifically:
- Is the paper sufficiently novel or is it just CPFT adapted to recent QA approaches? The general consensus is that QAID is indeed inspired by these works, but puts together a non-trivial combination of ideals to get achieve strong performance.
- Some questions associated with the ablation studies; it was generally argued that the rebuttal adequately addressed these concerns.